# SELF-GUIDED THINKING: ENABLING LLMS TO DECIDE WHEN TO THINK

## ABSTRACT

Large reasoning models improve performance on complex tasks by generating extended thought processes, but applying this approach uniformly to general user queries is computationally wasteful. Current solutions require complex multi-model systems or burden the user with manual controls. To address this, we introduce Self-Guided Thinking (SGT), a framework that enables a single model to learn to decide for itself when to think. SGT seamlessly integrates a lightweight penalty for deliberation into the Direct Preference Optimization (DPO) objective during the general alignment phase, teaching the model to balance performance with computational cost. Our experiments show that SGT learns a sophisticated, domain-adaptive policy. It achieves near-peak performance on general benchmarks while significantly reducing unnecessary thinking, and generalizes effectively to challenging out-of-distribution tasks by increasing its thinking where needed. On verifiable benchmarks, we find that while SGT preserves the model's reasoning capabilities, the general alignment stage does not substantially improve them over a fine-tuned baseline, suggesting the need for targeted in-domain training for further gains. Our ablations reveal that SGT teaches the model when to deploy a pre-existing capability, not how to reason from scratch; the policy's effectiveness is contingent on foundational knowledge from prior SFT and sufficient response length. Together, these findings demonstrate that an autonomous reasoning policy can be learned efficiently during general alignment, offering a practical path to deploy more economical and versatile models.

## 1 INTRODUCTION

Scaling inference-time compute has become the dominant paradigm for achieving state-of-the-art (SOTA) performance in domains where extended deliberation ("thinking") is critical, most notably in STEM tasks. Large Reasoning Models (LRMs) excel by trading this additional compute for higher accuracy, producing long "thoughts" before finalizing an answer (Xiang et al., 2025b; Comanici et al., 2025; OpenAI, 2025; Guo et al., 2025; Team et al., 2025). In real-world settings, the vast majority of user requests span a diverse spectrum of general tasks, including nuanced instruction-following, open-ended question answering, and role-playing etc, where this intensive thinking process often introduces significant latency without meaningfully improving user preference (Köpf et al., 2023; Zhao et al., 2024). Two primary strategies have emerged. One approach defers the decision to the user by engineering explicit "think" modes into their models (Anthropic, 2024). Another uses an external router to direct queries to either a powerful reasoning model or a lightweight, direct-answer model in a pool of existing LLMs (OpenAI, 2025; DeepSeek, 2025). Both strategies, however, have drawbacks: one burdens the user with choosing the correct mode, while the other requires training and maintaining complex systems.

Recent technical reports on SOTA open-source models provide insight into a full pipeline how a hybrid model can be built. The Qwen3 technical report, for instance, details a multi-stage post-training paradigm where models are first imbued with powerful reasoning abilities through supervised fine-tuning (SFT) and reasoning RL with verifiable rewards, followed by a "general RL" stage to align the model with broad user preferences for subjective, open-ended downstream tasks (Yang et al., 2025). This process, while creating a versatile model, has two notable outcomes. First, as their analysis shows, it results in a performance trade-off, where peak reasoning capability is degraded. Second, the final model relies on explicit user commands to switch between its "thinking" and "non-

thinking" modes. This raises a more fundamental question: can the decision of when to think be made autonomous rather than being offloaded to the user?

Our work investigates this possibility. We propose that the general RL stage is the ideal point to instill this capability by introducing Self-Guided Thinking (SGT), a method designed to be seamlessly integrated into the general alignment phase. By adding a lightweight penalty on reasoning tokens, our DPO-with-thinking-regularizer (DPO-tr) objective transforms the training goal: it simultaneously aligns the model with user preferences and teaches it to determine when to think. We shows that this approach is highly effective on UltraFeedback, SGT maintains near–"always-think" win rates while reducing the amount of thinking by 10–20%, and generalize to out-of-distribution (OOD) evaluations including Arena-Hard and Arena-Creative Writing. Furthermore, we find the model learns a domain-adaptive policy, as it apply "thinking" for complex reasoning tasks, while intelligently conserving computational resources on subjective tasks where direct responses suffice. Our ablations reveal that the policy's effectiveness on verifiable reasoning domains including math and coding is contingent on the model having both the foundational knowledge from prior SFT and a sufficient response length to execute its thought process. This demonstrates that an autonomous reasoning policy can be learned during the general alignment phase without additional cost, offering a practical path to deploy powerful models more economically.

## 2 RELATED WORK

**Efficient reasoning in large reasoning models.** Efficient reasoning in LRMs has emerged as a critical area of research, aiming to balance performance and computational overhead during inference. A common strategy is to augment reinforcement learning objectives with length penalties to encourage concise outputs, particularly in verifiable domains. Difficulty-aware approaches dynamically adjust the penalty based on estimated problem complexity, such as through adaptive length penalties (Xiang et al., 2025a) or explicit sampling strategies (Shrivastava et al., 2025). In contrast, difficulty-agnostic methods apply a uniform penalty to all reasoning traces (Team et al., 2025; Arora & Zanette, 2025), use techniques like clipping to shorten them (Hou et al., 2025), or allow users to set a maximum length (Aggarwal & Welleck, 2025). These length control methods focus on verifiable domains and operate on models that "think" by default. Their goal is to make this deliberation more efficient by regularizing total response length, rather than questioning if deliberation is necessary in the first place. In contrast, our SGT framework penalizes the act of thinking itself, not its length, teaching the model the more fundamental and flexible policy of whether to think at all.

**Hybrid reasoning** Hybrid reasoning, which combines fast, direct responses with deliberative thinking to optimize performance, is often implemented via router-based systems. These methods route queries to specialized "fast" or "reasoning" models (OpenAI, 2025;?), a flexible approach that requires maintaining multiple systems. The alternative is the single-model approach, where a unified model learns to adaptively switch modes. Existing methods typically offload the decision-making process: either to an internal learned policy that uses control tokens or difficulty predictors, often focused on verifiable domains (Jiang et al., 2025; Fang et al., 2025), or externally to the user via manual toggles (Yang et al., 2025; DeepSeek, 2025). Our Self-Guided Thinking (SGT) framework presents a distinct solution. By embedding a thinking regularizer directly into the general preference alignment phase, SGT is unique in that it learns an autonomous policy as an emergent outcome of aligning with broad user preferences.

## 3 METHOD

Our method for training a self-guided reasoning model consists of two key stages. The first is a Supervised Fine-Tuning (SFT) stage where we create a versatile base model by teaching it the format of both direct answers and responses via thinking. The second is the Self-Guided Thinking (SGT) stage where a selective policy is learned through online, iterative reinforcement learning from AI feedback (RLAIF).

### 3.1 Supervised Fine-Tuning for Hybrid Thinking

The first stage of our method prepares a hybrid model ($\pi_\theta$) capable of generating both direct answers and responses preceded by a reasoning trace. The goal is to instill the raw capability for hybrid thinking without a strong preference for either mode. To achieve this, we curate a dataset containing two distinct completion styles for each prompt: a direct response and a deliberated response where the thought process is enclosed in `<think> ... </think>` tags. To avoid inducing a quality bias, both response types are generated to be of comparable quality. The base model is then fine-tuned on this dataset using a standard SFT objective.

### 3.2 Self-Guided Thinking Behavior

To train the hybrid base model to learn when to think, we introduce our novel training method, Self-Guided Thinking (SGT). Due to lack of verifiable reward in open domains, we optimize the model's hybrid thought and response generation through RLAIF. Our approach builds upon Direct Preference Optimization (DPO) (Rafailov et al., 2023), modifying the standard objective to make the model not only aware of which response is better, but also when it is worth spending the extra compute on thinking.

#### 3.2.1 Preliminaries: Direct Preference Optimization (DPO)

DPO aligns a language model with a preference dataset $\mathcal{D} = \{(x, y_w, y_l)\}_i$, where for each prompt $x$, $y_w$ is the preferred (winning) response and $y_l$ is the dispreferred (losing) response. It directly optimizes a policy model $\pi_\theta$ to satisfy these preferences, using a fixed reference model $\pi_{\text{ref}}$, which is typically a supervised fine-tuned (SFT) version of the initial model.

The standard DPO objective function is:

$$L_{\text{DPO}}(\theta) = -\mathbb{E}_{(x,y_w,y_l) \sim \mathcal{D}} \left[ \log \sigma \left( \beta \log \frac{\pi_\theta(y_w \mid x)}{\pi_{\text{ref}}(y_w \mid x)} - \beta \log \frac{\pi_\theta(y_l \mid x)}{\pi_{\text{ref}}(y_l \mid x)} \right) \right]$$

Here, $\beta$ is a hyperparameter that controls the deviation from the reference policy. The core of this loss function is the term inside the sigmoid $\sigma(\cdot)$, which represents the difference in the log-probability ratios between the winning and losing responses. Intuitively, minimizing this loss encourages the policy $\pi_\theta$ to assign a significantly higher relative probability to the preferred response $y_w$ compared to the dispreferred response $y_l$.

#### 3.2.2 Online preference pair construction

We generate preference pairs through an online RLAIF process. For each prompt $x$ from our training distribution, we first generate a set of $k$ candidate responses, $Y = \{y_1, \ldots, y_k\}$, by sampling from our current policy $\pi_\theta(\cdot \mid x)$. These candidates are then scored by a preference (reward) model $r(y, x)$. A winning response $y_w$ and a losing response $y_\ell$ are selected from this set based on their scores, typically by choosing the highest- and lowest-scoring candidates:

$$y_w = \arg\max_{y \in Y} r(y|x) \qquad \text{and} \qquad y_\ell = \arg\min_{y \in Y} r(y|x).$$

Crucially, because the policy $\pi_\theta$ was prepared in the SFT stage to be capable of both response styles, the set of candidates $Y$ naturally contains a mix of direct and thinking-based answers. This ensures the resulting preference dataset $\mathcal{D} = \{(x, y_w, y_\ell)\}_i$ is populated with the hybrid pairs required for our DPO-tr objective.

#### 3.2.3 DPO with a Thinking Regularizer (DPO-tr)

The standard DPO objective is agnostic to the structure or computational cost of the responses; it only cares about which one is preferred. To encourage the model to learn an efficient reasoning policy, we introduce an explicit regularization term directly into the DPO loss function. We define an indicator function, $\mathbf{1}_{\text{think}}(y)$, which returns 1 if the response $y$ contains a deliberative reasoning trace (e.g., enclosed in `<think>...</think>` tags) and 0 otherwise.

Our modified objective, which we term **DPO with a thinking regularizer (DPO-tr)**, is defined as follows:

$$L_{\text{DPOtr}}(\theta \mid \mathcal{R}) = -\mathbb{E}_{(x,y_w,y_l)\sim\mathcal{D}} \left[ \log \sigma \left( \beta \log \frac{\pi_\theta(y_w \mid x)}{\pi_{\text{ref}}(y_w \mid x)} - \beta \log \frac{\pi_\theta(y_l \mid x)}{\pi_{\text{ref}}(y_l \mid x)} + \alpha \left( \mathbf{1}_{\text{think}}(y_w) - \mathbf{1}_{\text{think}}(y_l) \right) \right) \right] \tag{1}$$

The key innovation is the term $+\alpha \left( \mathbf{1}_{\text{think}}(y_w) - \mathbf{1}_{\text{think}}(y_l) \right)$, which adds a direct penalty or bonus to the log-odds based on the presence of thinking. The hyperparameter $\alpha$ controls the magnitude of the cost of thinking. Applying a non-zero $\alpha$ forces the model to learn that deliberation must be justified by a substantial quality improvement. Conversely, it provides a bonus when a direct response wins, rewarding the model for being efficient and concise. When both responses are of the same type, the term is zero, and the objective reverts to standard DPO.

## 4 EXPERIMENTAL SETUP

A central motivation for this work is to understand whether the utility of thinking is domain-dependent. To train and evaluate a model that can learn a selective reasoning policy, we require a data strategy that spans both complex, verifiable domains (math, coding and STEM) and broad subjective domains where user preference is the primary metric.

**Datasets** The goal of our SFT stage is to create a versatile base model with priors for both direct and thinking-based responses. To achieve this, we curate data from two distinct sources. For verifiable domains, we use the original thinking traces from OpenThoughts3 (Guha et al., 2025), filtering the dataset to 445k examples. We then generated corresponding direct answers for these prompts using `Qwen3-8B-Instruct` without thinking. To obtain subjective domain prompts we filter out math and coding prompts, which are redundant in OpenThoughts3, from the ServiceNow dataset (Madhusudhan et al., 2025). This resulted in 714k prompts. This curated data, sourced from ∼1.15 million unique prompts, is then compiled into two final SFT datasets. For this set, `Qwen3-8B-Instruct` is used to generate both the thinking and direct responses. This data is then compiled into two final SFT datasets: a direct dataset ($D_{\text{direct}}$) to train our non-thinking baselines, and a hybrid dataset ($D_{\text{hybrid}}$) (containing both response types for each prompt) to prepare our model for the RLAIF stage.

**RLAIF Dataset.** For the RLAIF alignment stage, our focus shifts from teaching the capability of reasoning to teaching the policy of when to apply it based on broad user preferences. For this, we use the UltraFeedback dataset, which we term $D_{\text{RLAIF}}$, a large-scale collection of user-assistant conversations across 21 diverse and primarily subjective domains. We use 59.8k prompts for training and a held-out set of 4.2k prompts for in-domain evaluation.

**Training Details.** Our training begins with `Qwen3-4B` base and consists of two main stages. In the Supervised Fine-Tuning (SFT) stage, we create two initial models. The **Direct SFT** model is trained $D_{\text{direct}}$. The **Hybrid SFT** model, trained on $D_{\text{hybrid}}$, serves as the foundation for our method. Both models are trained for 2 epochs. In the second stage, we apply preference tuning on $D_{\text{RLAIF}}$. The **Direct DPO** model is created by further training the Direct SFT model with standard DPO. Our primary **SGT** model is created by training the Hybrid SFT model with our DPO-tr objective. This RLAIF training is conducted via an online DPO process for 9 steps, where for each prompt, a reward model, `Athene-RM-8B` (Frick et al., 2024), scores four generated candidates to select the winning and losing pair for the update. We chose this reward model as it was top-ranked on the Reward Bench Leaderboard at the start of our experiments[1]. Compute details can be found in Section A.1.

**Hyperparameters and Ablations.** For our main experiments, all models were trained with a maximum response length of 16K. We test a range of $alpha$ values (0, 1.2, 1.3, 2) to analyze the effect of the thinking penalty. To test the boundary conditions of our method, we also run two ablation studies: one with a reduced 8K response length, and another using a Hybrid SFT model trained on an SFT dataset with no STEM data.

---

[1]https://huggingface.co/spaces/allenai/reward-bench

**Evaluation.** We evaluate our models across three categories of benchmarks. For in-domain subjective evaluation, we use a held-out test set of 4.2K prompts from UltraFeedback. For out-of-distribution (OOD) subjective domains, we use Arena-Hard and Arena-Creative Writing to test for generalization. Finally, for verifiable evaluation, we test on AIME 24-25, OlympiadBench, and LiveCodeBench to measure objective correctness on complex reasoning tasks. The evaluation protocol differs by benchmark. For the UltraFeedback test set, we use GPT4.1-nano as an LLM judge to determine the winning response. For Arena-Hard and Arena-Creative Writing, we follow the official pairwise evaluation protocol from the benchmark (Li et al., 2024). On the verifiable benchmarks, we measure Pass@1 accuracy. Our primary analysis focuses on two controlled comparisons: Direct SFT vs. Hybrid SFT and Direct DPO vs. SGT, as these pairs share the same initial training conditions and allow us to isolate the effects of our method.

## 5 RESULTS

### 5.1 HYBRID THINKING IN SUBJECTIVE DOMAINS

Table 1: Evaluation in subjective domains. Win is win rate (%) vs the same baseline per benchmark; Think is think rate (%). Win rate are computed against Direct SFT for UltraFeedback, o3 for Arena-Hard and Gemini-2.0-Flash for Arena-Creative Writing.

| Model | UltraFeedback | | Arena-Hard | | Arena-Creative Writing | |
|---|---|---|---|---|---|---|
| | Win | Think | Win | Think | Win | Think |
| Direct SFT | 50 | 0 | 7.7 | 0 | 7.3 | 0 |
| Hybrid SFT | 52.6 | 59.4 | 5.6 | 55 | 9 | 50 |
| Direct DPO | 53.2 | 0 | 7.4 | 0 | 9.8 | 0 |
| SGT | **56.3** | 81.4 | **20** | 75 | **27** | 74 |

**Effective hybrid thinking emerges as a result of RLHF with thinking cost.** Our results in Table 1 show that Self-Guided Thinking (SGT) learns a highly effective, selective reasoning policy that significantly outperforms baseline models, especially on challenging domains. On the in-domain UltraFeedback task, we observe that both the naive Hybrid SFT and the final SGT model achieve comparable gains over their non-thinking counterparts. However, the limitations of a naive approach by simply enabling a "think" mode yields inconsistent outcomes in OOD tasks. First, the Hybrid SFT model underperforms the Direct SFT model on the reasoning-intensive Arena-Hard benchmark (5.6% vs. 7.7% win rate). Further preference tuning without a thinking mechanism also fails to improve performance on these OOD reasoning tasks. In contrast, SGT dramatically boosts performance across the board. On Arena-Hard, which contains difficult math and coding problems, SGT improves the win rate nearly threefold over the best baseline (from 7.7% to 20%). It achieves a similar leap on Arena-Creative Writing (from 9.8% to 27.0%). This performance gain is directly linked to its learned policy: SGT activates its reasoning capabilities on 74-75% of prompts in these challenging domains. This demonstrates that by introducing a cost for deliberation during RLHF, SGT learns to strategically apply "thinking" where it has the most impact, achieving superior generalization and performance.

**The Utility of Deliberative Reasoning is Highly Domain-Dependent.** As illustrated in Figure 1, our analysis of the in-domain UltraFeedback test set shows that the model learns a nuanced, domain-specific reasoning policy. Thinking provides a significant advantage on tasks requiring structured planning or complex instruction-following, such as Conversational Dialogue and Reasoning and Problem Solving. Conversely, its utility diminishes in more subjective domains like Art and Direct, where a direct, "System-1" style response suffices. Perhaps the clearest evidence of this selective policy is the model's behavior on the Math and Calculations tasks within UltraFeedback. Here, the model correctly infers the low complexity of the problems and adaptively reduces its thinking rate, demonstrating an ability to balance cost against expected performance. This reduction in think rate on simple, in-domain math is thrown into sharp relief when tested on a challenging OOD benchmark. On Arena-Hard, a domain composed of difficult math and coding problems, SGT achieves its most dramatic performance gain, nearly tripling the win rate over the baseline (Table 1). The

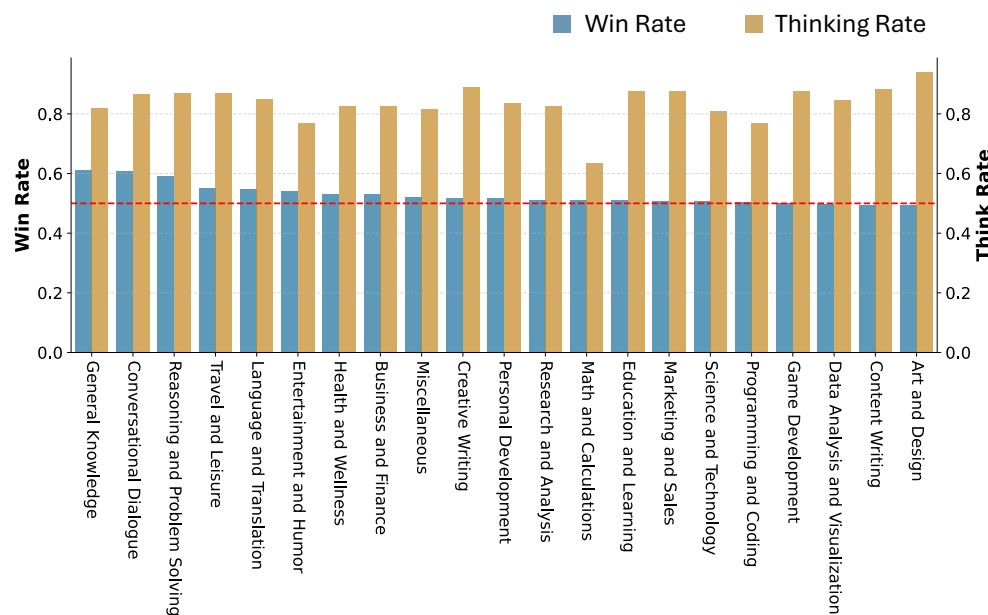

Figure 1: **SGT Performance by Domain.** Win and think rate of the SGT model against the direct response DPO baseline after 9 steps of training $\alpha = 1.3$.

stark contrast-between its low thinking rate on simple math and its high (75%) rate on complex problems-proves that SGT learns a sophisticated policy. It assesses not just the domain, but the inherent difficulty of a problem to strategically apply its reasoning capabilities. This principle of superior generalization is also evident in subjective domains. For instance, while the SGT model shows only a marginal win rate improvement in Creative Writing on the in-domain UltraFeedback test set, the gap widens significantly on the Arena-Creative Writing benchmark (Table 1). We hypothesize that this is because the standard Direct DPO model overfits to the stylistic preferences within the UltraFeedback domain. In contrast, SGT's thinking process enables it to learn a more abstract and generalizable strategy for creative writing tasks, allowing it to maintain high performance even when the style and nature of the prompts shift.

## 5.2 PERFORMANCE ON VERIFIABLE TASKS.

Table 2: Verifiable results comparing our SGT model against the Direct Baseline. For math benchmarks, we report Pass@1 accuracy (%) estimated with 64 samples.

| Model | AIME 24 | | AIME 25 | | OlympiadBench | | LiveCodeBench | |
|---|---|---|---|---|---|---|---|---|
| | Acc | Think | Acc | Think | Acc | Think | Acc | Think |
| Direct SFT | 16.6 | 0 | 14.3 | 0 | 37.5 | 0 | 12.4 | 0 |
| Hybrid SFT | 23.8 | 39 | 21.6 | 37 | 42.6 | 38 | 19.2 | 42 |
| Direct DPO | 15.7 | 0 | 15.7 | 0 | 38 | 0 | 13.6 | 0 |
| SGT | **25.6** | 39 | **22.3** | 38 | **43.2** | 40 | **19.5** | 42 |

To understand how the RLAIF stage impacts specialized STEM/code/math reasoning, we now turn to our suite of verifiable benchmarks. The results in Table 2 confirm that while our SGT policy preserves existing capabilities, it does not significantly enhance them without targeted training. Consistent with our findings in subjective domains, the Direct DPO model shows no improvement over the Direct SFT baseline, confirming that general preference tuning does not impart the necessary skills for these tasks. Interestingly, the naive Hybrid SFT model provides a substantial performance lift, suggesting that for these highly complex problems, any form of step-by-step reasoning is beneficial. However, while our SGT model maintains a slight performance edge, it does not meaningfully im-

prove upon the Hybrid SFT baseline. We attribute this to the fact that its think rate on these problems remains comparable to that of the Hybrid SFT model (Table 2). This suggests that the calibration learned during the general RLHF stage does not transfer to these specialized domains; the model doesn't learn to increase its deliberation for these difficult problems. This highlights a key limitation: to advance performance on complex STEM tasks, targeted in-domain training is required.

### 5.3 Effects of Hyperparameters and Ablation

Having established the effectiveness of SGT, we now investigate the necessary conditions for this policy to emerge by directly addressing our third research question. To understand the policy's dependence on initial training and hyperparameters, we conduct a series of ablations. We analyze the sensitivity to the thinking penalty, $\alpha$, the effect of a constrained max response length, and the necessity of prior STEM knowledge by training a model on an SFT dataset with no STEM problems.

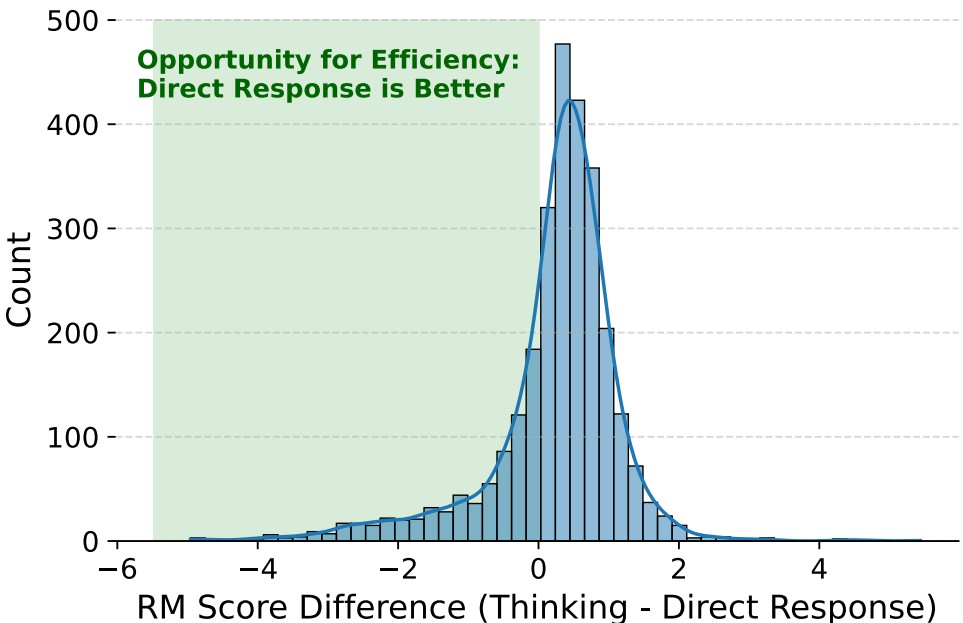

Figure 2: **Distribution of reward score differences between 'thinking' and 'direct' responses from the initial SFT model.** Responses after thinking usually score higher than direct responses based on the reward model, while the negative tail highlights prompts where a direct response is superior, justifying a selective policy

**A balanced $\alpha$ is critical for an effective performance-cost trade-off.** First, the choice of the thinking penalty, $\alpha$ is critical. As shown in Figure 3, $\alpha$ directly controls the trade-off between performance (Win Rate) and computational cost (Thinking Rate). An $\alpha = 0$ policy learns to "always-think" maximizing its reward by following the reward model's general preference for responses after thinking. Figure 2 illustrate the distribution of reward scores is positively skewed before general RL, confirming the reward model generally assigns higher scores to the higher-quality answers produced through thinking. However, the crucial negative tail of the distribution identifies prompts where a direct response is superior, demonstrating that a fixed "always-think" policy is suboptimal and creates a clear opportunity for a more efficient, selective policy. Conversely, setting the penalty too high ($\alpha = 2$) aggressively suppresses thinking to the model's detriment, causing its win rate to collapse below the non-thinking baseline. The success of SGT lies in finding a balanced $\alpha$ (1.2, 1.3) that that learns to forgo thinking only when the quality drop is minimal. These models effectively learn to identify and act on the efficiency opportunities presented by the negative tail, confirming that our DPO-tr objective can successfully teach a model to think when it is beneficial.

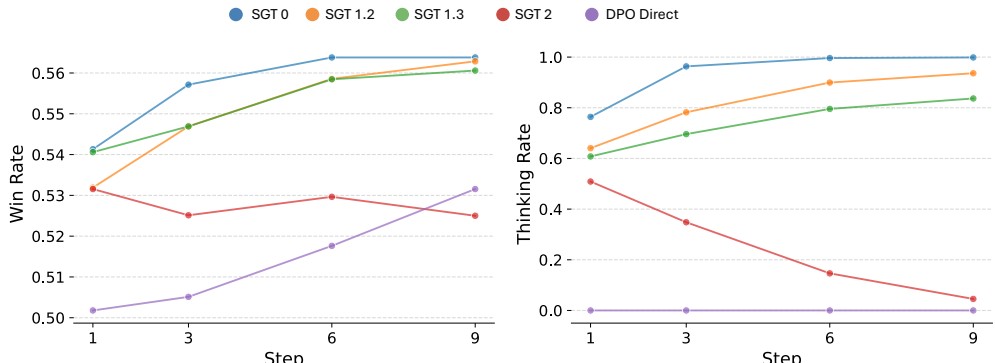

Figure 3: **SGT Learns an Efficient Reasoning Policy.** Win rate (left) and thinking rate (right) for models trained with varying thinking penalties ($\alpha$) on the UltraFeedback test set. SGT models ($\alpha = 1.2, 1.3$)) learn to reduce their thinking rate (cost) without a significant drop in performance compared to the "always-think" model ($\alpha = 0$).

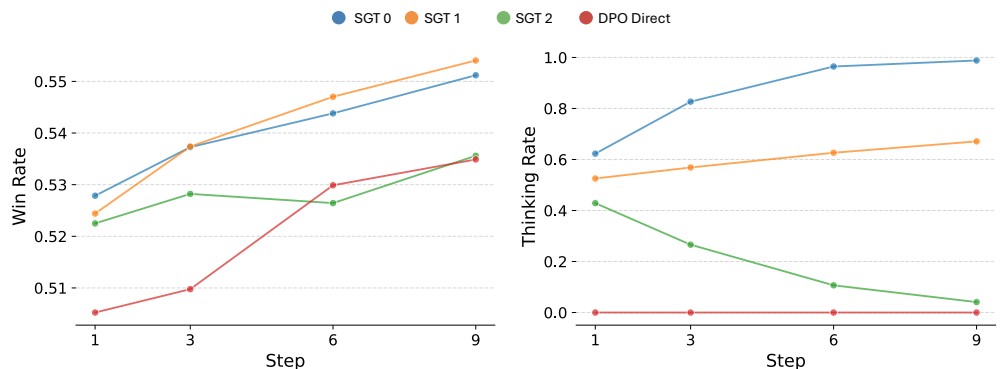

Figure 4: **GT Learns an Efficient Reasoning Policy without STEM SFT data and 8K max response length.** Win rate (left) and thinking rate (right) for models trained with varying thinking penalties ($\alpha$) on the UltraFeedback test set. SGT models ($\alpha = 1$)) learn to reduce their thinking rate (cost) without a significant drop in performance compared to the "always-think" model ($\alpha = 0$).

Table 3: Verifiable results comparing our SGT model against the Direct Baseline. For AIME 24-25, we report Pass@1 accuracy (%) estimated with 64 samples. OlympiadBench and LiveCodeBench both used 8 samples.

| Model | AIME 24 | AIME 25 | OlympiadBench | LiveCodeBench |
|---|---|---|---|---|
| Direct DPO (16K) | 15.8 | 15.7 | 38.2 | 13.6 |
| SGT-0 (16K) | 35.5 | 30.1 | 48.3 | 19.5 |
| SGT-1.2 (16K) | 27.6 | 23.4 | 44.4 | 19.9 |
| SGT-1.3 (16K) | 25.6 | 22.3 | 43.2 | 19.5 |
| SGT-2 (16K) | 17 | 17 | 40 | 14.2 |
| Direct Baseline (8K) | 15.7 | 15.7 | 38 | 13.4 |
| SGT-1.2 (8K) | 17.6 | 14.8 | 37.7 | 13 |
| Direct Baseline (8K, No STEM) | 16.6 | 12.9 | 38.5 | 13 |
| SGT-1.2 (8K, No STEM) | 17.1 | 13.8 | 38 | 13.4 |

**The effect of max response length during general RL.** The extended reasoning process in "think mode" can significantly increase response length. While a higher length cap during training allows

the model to explore more comprehensive reasoning paths, it also increases the computational budget. To understand the impact of a more constrained budget, we trained a new suite of models with the max response length reduced from 16K to 8K. On the general-domain UltraFeedback test set, a constrained budget primarily affects the learning dynamics rather than the final performance (Figure 4). The "always-think" model ($\alpha = 0$) converges to a 100% thinking rate more slowly, taking 6 steps instead of 3. For a balanced penalty ($\alpha = 1$), the model still achieves a final win rate comparable to the always-think model. This might suggest that a shorter response length is sufficient for general RLHF. However, Table 3 reveals a different story for specialized domains. Constraining the response length to 8K causes a catastrophic drop in performance on the verifiable STEM benchmarks, with the SGT-1.2 (8K) model's accuracy on AIME 24 collapsing from 28% to 15%. This is because complex problems require longer reasoning traces, and a constrained budget prevents the model to learn necessary strategies to solve them. These results indicate that while a shorter context may suffice for general-domain alignment, a longer max response length during training is critical for preserving high-level reasoning capabilities in STEM.

**Alation of STEM SFT data.** Our final and most critical ablation tests the necessity of prior domain knowledge by training a model exclusively on a non-STEM SFT dataset. The results from the UltraFeedback test set (Figure 4) show that, even with this limited prior, the model still learns a selective policy—it successfully trades win rate for a lower thinking rate as $\alpha$ increases. However, this learned policy proves to be entirely superficial when applied to actual STEM problems. As shown in Table 3, the SGT-1.2 (8K, No STEM) model performs no better than its non-thinking Direct Baseline counterpart on verifiable benchmarks. For instance, both models score 17% on AIME 24, and the SGT model achieves no reliable gains on AIME 25. This is a crucial finding. It proves that SGT is not a mechanism for creating reasoning ability from scratch; instead, it is a powerful method for learning to selectively deploy a pre-existing capability. Foundational knowledge from domain-specific SFT is a necessary prerequisite for the SGT policy to be effective on challenging tasks. To test the preconditions for this policy, we conducted a critical ablation study: we trained a model without any prior STEM SFT data and with a reduced 8K maximum response length. The results in Figure 4 show that, with smaller response length, the model still learns a selective policy. However, this learned policy is superficial. Instead, it is a powerful method for learning to selectively deploy a pre-existing capability. Both foundational knowledge from domain-specific SFT and sufficient context length to execute complex reasoning are necessary prerequisites for the learned policy to be effective on challenging tasks.

## 6 DISCUSSION AND LIMITATIONS

In this work, we introduced Self-Guided Thinking (SGT), a method that teaches a single model to autonomously decide when to engage in costly thinking by integrating with DPO-tr. Our findings demonstrate that SGT learns a domain-adaptive policy; this alignment phase teaches the model when to deploy a pre-existing thinking capability. The effectiveness of this policy is therefore contingent on the model's foundational SFT, and we find that while our general alignment preserves specialized STEM skills, targeted in-domain training would be required to further enhance them. Ultimately, SGT offers a practical path toward deploying more versatile and economical models that can efficiently govern their own computational resources. Our work has several limitations. First, the SGT policy's effectiveness is capped by the model's foundational reasoning ability from the SFT stage; future work could combine our general alignment method with targeted RL to jointly enhance and manage these skills. Second, our thinking penalty ($\alpha$) is global, and a more sophisticated approach could learn a dynamic penalty that adapts to the context, leaving space for follow-up work.

### AUTHOR CONTRIBUTIONS

If you'd like to, you may include a section for author contributions as is done in many journals. This is optional and at the discretion of the authors.

### ACKNOWLEDGMENTS

Use unnumbered third level headings for the acknowledgments. All acknowledgments, including those to funding agencies, go at the end of the paper.

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

## A APPENDIX

### A.1 ADDITIONAL EXPERIMENTAL DETAILS

Our experiments were conducted on of two individual nodes, each equipped with 8 NVIDIA H100 GPUs. For the SFT stage, models were trained for 2 epochs, with the Hybrid SFT model taking approximately 3 days to complete. The subsequent RLAIF training was conducted for 9 steps over the $D_{\text{RLAIF}}$ dataset using a batch size of 4400 and a DPO hyperparameter of $\beta = 0.1$. During this online DPO process, we sampled 4 candidate responses per prompt at a temperature of 0.7 and tested a range of $\alpha$ values (0, 1.2, 1.3, 2). A full 16K context SGT run took ∼22 hours, while the 8K context and Direct DPO models each took ∼12 hours. For all final evaluations, the maximum response length for all models was set to 16K.

### A.2 FINEGRAINED DOMAIN EVALUATION

The comparison between the domain-level performance of our main models (Figure 6) and that of our ablation models (Figure 5) powerfully illustrates the necessary preconditions for SGT to be effective. In our main experimental setting, the fully-resourced models show strong and consistent win rate improvements across nearly all 21 domains as training progresses, including areas like Math and Calculations and Programming and Coding. In stark contrast, the ablation models—trained without STEM SFT data and with a constrained 8K response length—fail to achieve meaningful gains in these same critical domains, even when configured to "always-think" ($\alpha = 0$). This visual evidence demonstrates that the selective reasoning policy learned by SGT is not superficial; it is highly effective, but its success on challenging tasks is contingent on the model having both the pre-existing knowledge and the sufficient context length required to execute complex deliberation.

### A.3 EXAMPLES OF USEFUL THINKING

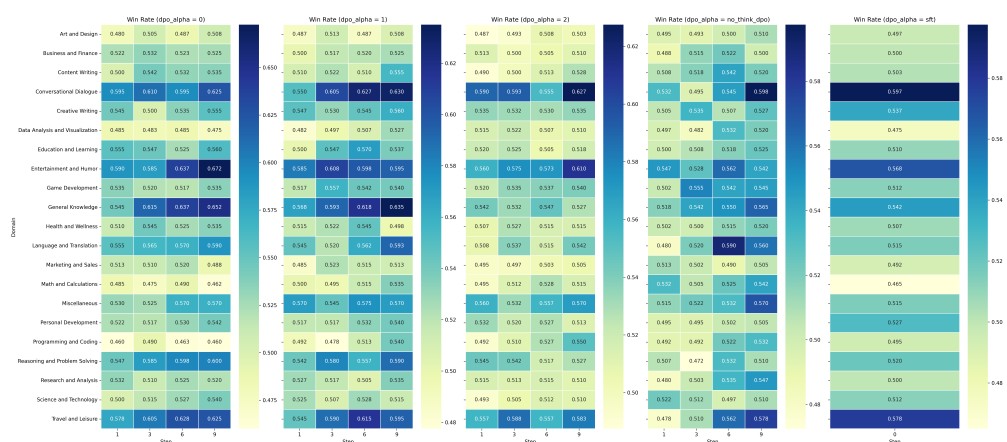

Figure 5: Domain-level win rate evolution for ablation models. Each heatmap shows the win rate on the UltraFeedback test set across 21 domains and 9 training steps for a specific model configuration trained without STEM SFT data and with an 8K maximum response length.

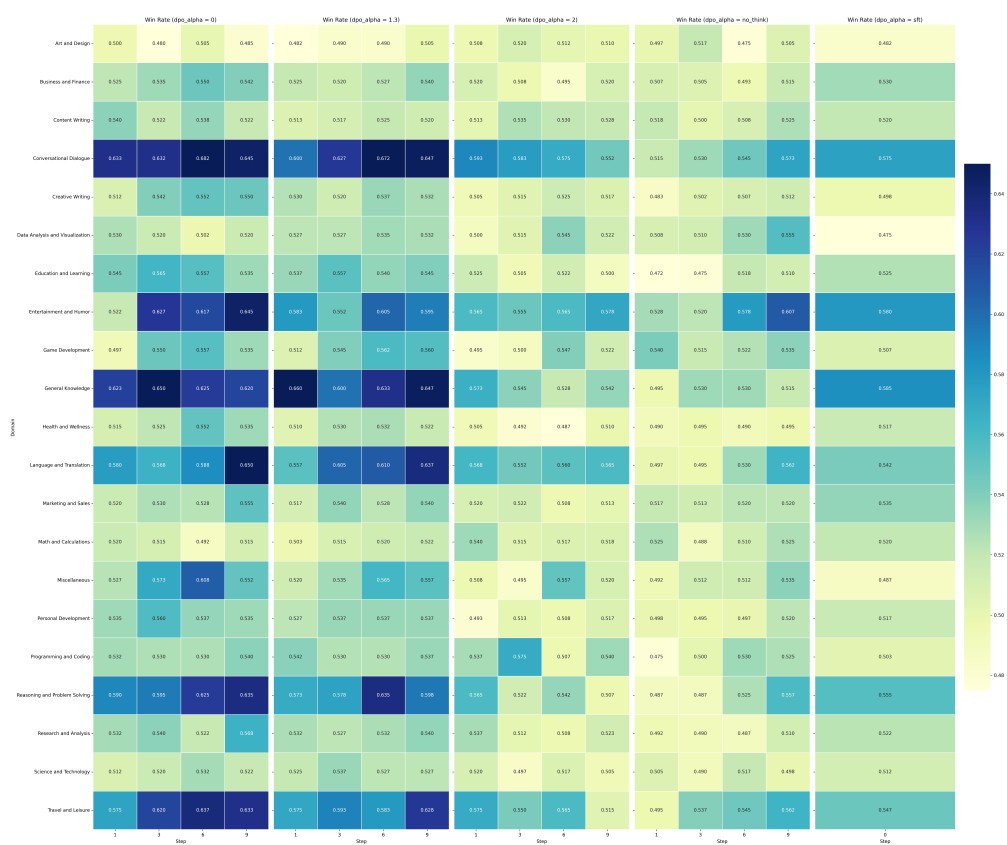

Figure 6: Domain-level win rate evolution for ablation models. Each heatmap shows the win rate on the UltraFeedback test set across 21 domains and 9 training steps for a specific model configuration trained with STEM and subjective domain SFT data and with an 16K maximum response length.

| Prompt | Domain | Helpful Thinking |
|---|---|---|
| Could you pretend to be my friend for this conversation? | Conversation Dialogue | `<think>` ... First, I need to establish a friendly and approachable tone. Maybe start with a casual greeting to make them feel comfortable. I should ask how they're doing to show genuine interest. It's important to listen and respond empathetically. Let me make sure my responses are open-ended so they can share more. I should avoid any technical jargon and keep the language simple and conversational. Also, I need to be mindful of their possible needs... `</think>` |
| Write a story about a person who can change their own weight and density. | Creative Writing | `<think>`...Let me start by brainstorming some ideas. First, I need to figure out how the ability works. Maybe the protagonist can adjust their weight and density at will, which could have various effects. Let me think about the implications. If someone can change their weight, they could float, become heavier to sink, or even manipulate objects. But how does that tie into a story? I should create a relatable protagonist. Maybe someone who discovers this ability accidentally. Let's say they're a teenager, which makes the story more engaging... `</think>` |

Table 4: Examples of prompts with "Helpful Thinking" responses in Conversation Dialogue and Creative Writing domains.

