# OpenReview forum: "Self-Guided Thinking: Enabling LLMs to Decide When to Think"
_ICLR.cc/2026/Conference — ICLR 2026 Conference Withdrawn Submission_

### Official Review · Reviewer_hKeW · 2025-10-28

**Soundness:** 2
**Presentation:** 2
**Contribution:** 2
**Rating:** 4
**Confidence:** 3

**Summary:**

This paper introduces Self-Guided Thinking (SGT) method, enabling models to autonomously decide when to activate thinking mode in response to prompts. Experiment results show that SGT can learn complex domain-adaptive strategies.

**Strengths:**

- This paper demonstrates sound research motivation, as enabling models to autonomously determine when to activate their thinking mode holds practical significance for constructing more efficient models.

- The proposed 2-stage pipeline is quite intuitive, and the proposed loss function looks reasonable to me.

**Weaknesses:**

**Missing Important Baselines**: There is an important baseline that is training the Hybrid SFT model with standard DPO (i.e., setting $\alpha=0$ in the new proposed loss). This is a critical baseline to demonstrate the usefulness of the newly designed loss. Without such a baseline, it's hard to tell if the SGT model makes more efficient thinking strategies without much performance loss compared to the model with no such deliberation.

**Experiment Implementation**: Appendix A.1 does not present the training hyperparameters for the SFT stage. The choice of an extremely large batch size of 4400 for DPO training, coupled with executing only 9 DPO steps, appears quite peculiar.  How was such a large batch size determined? Was hyperparameter search conducted, or was it set based on some prior research work? This batch size choice seems somewhat unusual to me. If hyperparameter settings are unreasonable during the experiments, it becomes difficult to draw meaningful conclusions in the end.


**Presentation issues**: There are several typos and formatting issues.
- Line 91: There is a question mark in the citation.
- Line 202: "trained $D_\text{direct}$" --> "trained on $D_\text{direct}$"
- Line 263: "Art and Direct" --> "Art and Design"
- Line 478-485 can be moved.

**Questions:**

- In Appendix A.1, the paper mentions that RLAIF training uses a batch size of 4400. How was such a large batch size determined? Was hyperparameter search conducted, or was it set based on some prior research work? This hyperparameter choice seems somewhat unusual to me.
- This paper demonstrates the impact of the chosen $\alpha$ value on the UltraFeedback test set. Since UltraFeedback has been used as the test set, was a development set also employed to ultimately determine which $alpha$ value to adopt?

---

### Official Review · Reviewer_VG6i · 2025-10-30

**Soundness:** 1
**Presentation:** 2
**Contribution:** 1
**Rating:** 2
**Confidence:** 4

**Summary:**

This paper proposes a framework called Self-Guided Thinking, SGT, aimed at solving the problem of the waste of computational resources when reasoning. Different from existing methods which need complicated multi-model system, SGT is able to have a single model to learn to when it needs to think deeply. It introduces a lightweight regularization term in DPO's objective function to teach the model to balance the effect and the cost. The experiment results show that SGT can learn a strategy applied to various fields. For instance, it thinks more about hard tasks while thinking less about easy ones.

**Strengths:**

1. The experiment and evaluation system is comprehensive, including in-domain subjective tasks and out-of-domain objective ones, which shows the framework's effectiveness and generalization ability.
2. The paper also discusses the essence of SGT's power. It points that SGT teaches the model when to selectively deploy a pre-existing capability.

**Weaknesses:**

1. Limited backbone diversity.
 The main experiments only include Qwen3 as the backbone model. The authors should consider incorporating more long reasoning models to better demonstrate the general effectiveness of the proposed method. Moreover, it would be valuable to evaluate the method’s performance across different parameter scales.
2. Lack of key baselines.
 The paper misses comparisons with many relevant existing works, such as [1][2]. These should be included as baseline models to ensure a fair evaluation. In addition, the related work section does not clearly explain the key differences between the proposed method and these prior approaches.
3. Concerns about fairness and reliability of the reported results.
 The experimental details are not described clearly. For example, the paper states “For math benchmarks, we report Pass@1 accuracy (%) estimated with 64 samples,” but it is not fully explained how this estimation is performed. Furthermore, Qwen3 itself provides both think and no-think modes, how does the proposed SGT model perform compared to these original settings? The authors should report the backbone model’s performance on all benchmarks for a fair comparison. Additionally, the final results of SGT appear to differ significantly from the accuracies reported on the official Qwen3, raising concerns about result validity.
4. Numerous typographical and presentation issues.
 There are several typos and formatting problems throughout the paper. For instance, in the Abstract, the sentence beginning with “We shows that this approach…” contains a subject–verb agreement error and should be corrected to “We show…”. In Section 5.3, the subheading “Alation of STEM SFT data” should be “Ablation…”. Equation (1) also extends beyond the main text width and needs proper formatting.

[1] AdaptThink: Reasoning Models Can Learn When to Think

[2] ARM: Adaptive Reasoning Model

**Questions:**

see weakness

---

### Official Review · Reviewer_PYQF · 2025-10-30

**Soundness:** 1
**Presentation:** 3
**Contribution:** 1
**Rating:** 2
**Confidence:** 2

**Summary:**

This paper proposes an approach to train models to learn how to switch between thinking and non-thinking using a modified version of DPO with a regularizer that penalizes overthinking. It is measured on reasoning and non-reasoning tasks.

**Strengths:**

Knowing when to think and not think is a useful skill for frontier models.

**Weaknesses:**

Sec 5.1: I think you are missing two baselines of: (1) only training on thinking data at SFT stage (think 100%); and (2) DPO on top of that. Otherwise these results are just showing that more thinking helps right now?
There should also be baselines of combining the 2 separate models (think, no-think) which is a standard approach, as you say in the introduction.

Further, SOTA models use online RL training for CoT (and in general for responses), so it’s not clear how any results from SFT+DPO generalize to that case? I think when training CoT with RL it would be hard to put a regularizer on CoT length on that, as those techniques try to learn to make the reasoning longer (test time scaling). So I suppose a stage after that as you do might still make sense...

Dataset:
“To achieve this, we curate a dataset containing two distinct completion styles for each prompt: a direct response and a deliberated response where the thought process is enclosed in ... tags. To avoid inducing a quality bias, both response types are generated to be of comparable quality.” :    – this is surprising as generally responses after think would be expected to be better, so seems this dataset is quite unusual – probably needs more explanation how that could be the case?


The precise data construction is hard to understand.
For example: “...then compiled into two final SFT datasets. For this set, Qwen3-8B-Instruct is used to generate both the thinking and direct responses. This data is then compiled into two final SFT datasets: a direct dataset..”  –  so, you have 2 final SFT datasets? This is confusing.


grammar issue:
- “as it apply “thinking” for complex reasoning”

**Questions:**

See weaknesses.

---

### Official Review · Reviewer_fqQM · 2025-11-01

**Soundness:** 3
**Presentation:** 2
**Contribution:** 2
**Rating:** 4
**Confidence:** 3

**Summary:**

This paper introduces Self-Guided Thinking, an approach designed to address the computational inefficiency of Large Reasoning Models (LRMs) that apply costly, extended reasoning to all queries. The core contribution is a modified Direct Preference Optimization objective, "DPO-with-thinking-regularizer" (DPO-tr), which integrates a lightweight penalty for generating thinking tokens directly into the general alignment phase. This penalty teaches the model to autonomously balance response quality with computational cost, learning a selective policy on when to think. Experiments demonstrate that SGT trains a model to be domain-adaptive, significantly reducing unnecessary deliberation on simpler subjective tasks while strategically increasing its thinking on challenging, out-of-distribution reasoning problems.

**Strengths:**

1. The paper's core idea is simple and easy to implement. By introducing a single regularization term to the standard DPO objective, the method avoids the need for complex architectural changes, multi-stage training pipelines, or auxiliary models. This makes SGT a highly practical approach.
2. A significant strength of this work is the model's demonstrated ability to generalize its learned policy. The SGT model learns a nuanced, domain-adaptive policy rather than a simple, universal heuristic. More impressively, it successfully applies this policy to OOD benchmarks.

**Weaknesses:**

1. The technical contribution is not very clear. The proposed method adds a regularization term to the DPO objective, but there already exist several works that introduce penalty terms in on-policy optimization frameworks [1,2,3]. The main distinction appears to be that this method specifically targets DPO, which may limit its novelty.
2. Building on the above point, it would be beneficial to include more baselines for comparison. In particular, it is unclear whether a DPO-based approach performs better than PPO- or GRPO-based alternatives.
3. The presentation could be improved. For instance, including a framework figure to illustrate how the dataset is constructed and utilized during training would make the methodology clearer.
4. The method does not rely on explicit rules or verifiable rewards, which is great for flexibility. However, it raises a concern about whether this design might lead to incorrect or inconsistent responses in the training data.

**Questions:**

See weaknesses.

---

### Note · Authors · 2026-01-20

I have read and agree with the venue's withdrawal policy on behalf of myself and my co-authors.